# Universal Design for Learning for Children with ADHD

**DOI:** 10.3390/children10081350

**Published:** 2023-08-04

**Authors:** Alessandro Frolli, Francesco Cerciello, Clara Esposito, Maria Carla Ricci, Rossana Pia Laccone, Fabio Bisogni

**Affiliations:** 1Disability Research Centre of Rome University of International Studies, 00147 Rome, Italy; francesco.cerciello@unint.eu (F.C.); clara.esposito@unint.eu (C.E.); rossana.laccone@gmail.com (R.P.L.); fabio.bisogni@unint.eu (F.B.); 2FINDS—Italian Neuroscience and Developmental Disorders Foundation, 81040 Caserta, Italy; m.ricci@unint.eu

**Keywords:** Universal Design for Learning: learning, ADHD, hyperactivity, impulsivity, teaching

## Abstract

Attention Deficit–Hyperactivity Disorder (ADHD) is a psychiatric condition that shows developmentally inappropriate levels of inattention, hyperactivity, or impulsivity. Symptoms begin at a young age and usually include a lack of attention, poor concentration, disorganization, difficulty completing tasks, forgetfulness, and losing things. It is important to diagnose and treat the disorder at a young age so that the symptoms do not persist into adulthood and cause other comorbid conditions. Learning difficulties, motor impairment, anxiety, or depressive disorders may occur with this condition. To improve the academic careers of children with ADHD, we focused on a specific innovative educational approach (Universal Design for Learning) that could improve basic learning skills (reading, writing, and arithmetic skills) to prevent or manage any learning difficulty that could occur with ADHD. The Universal Design for Learning is an individualized approach that combines current neuroscientific knowledge, creating personalized teaching based on the strengths and weaknesses of the student. The goal of this study is to analyze the impact that this approach has on basic learning abilities. We found that both interventions led to improvements in test performance, indicating that interventions were necessary to enhance reading, writing, and arithmetic skills. Furthermore, the group that received an educational intervention based on Universal Design for Learning demonstrated a more significant improvement in these areas. Additionally, we propose that the set of techniques implemented by teachers in the classroom helped children to read, write, and perform math tasks correctly and more fluently.

## 1. Introduction

Attention Deficit–Hyperactivity Disorder (ADHD) is a psychiatric condition that has long been acknowledged as having an impact on children’s functioning. Individuals affected by this disorder exhibit levels of inattention, hyperactivity, or impulsivity that are not developmentally appropriate. Symptoms typically manifest at an early age and commonly include a lack of attention, poor concentration, disorganization, difficulty completing tasks, forgetfulness, and misplacing items. Diagnosing and treating the disorder early in life is crucial to prevent the persistence of symptoms into adulthood, which can lead to the development of other comorbid conditions [1]. ADHD is a prevalent psychiatric disorder that primarily affects children and adolescents. Due to its diverse nature as a developmental disorder, its exact causes remain unclear, leading to biased and extensive diagnostic assessments involving traditional clinical interviews and behavior evaluations [2,3]. In children, ADHD is characterized by hyperactivity, which results in an inability to control impulses and attention deficits that minimally impact their social engagement and daily activities. As individuals with ADHD transition into adulthood, they may encounter challenges related to time management, organization, goal-setting, and maintaining employment. These difficulties can further complicate relationships, lower self-esteem, and potentially lead to addiction. ADHD is the most common neurodevelopmental disorder in childhood, affecting approximately 4–8% of children worldwide [2]. The prognosis of ADHD is often complicated by the presence of comorbidities, and impairments may worsen during adolescence or adulthood [4,5]. The treatment for ADHD is typically multimodal, involving a combination of approaches, such as medication, psychoeducation, and psychological interventions [6]. However, the current multimodal approach has certain limitations [7]. One significant challenge is motivating individuals with ADHD to engage consistently in their treatment [8]. Additionally, psychotherapies can be costly [9], and there is often a high risk of dropout [10,11]. ADHD is associated with psychosocial functional impairment and significantly reduced subjective health-related quality of life [12,13]. Children with ADHD are about four times less likely to graduate from university compared to their peers and tend to have a lower socioeconomic status on average [14]. They often experience conflictual relationships with parents, siblings, peers, and partners [12,14]. Their risk of delinquency is increased by a factor of two–three [14,15]. Approximately 75% of individuals with ADHD have an additional mental disorder, and about 60% have multiple comorbid mental disorders, which can impact prognosis and require specific therapeutic interventions [16]. Specifically, learning difficulties (e.g., in reading, writing, and arithmetic), motor impairments, anxiety disorders, and oppositional defiant behavior may occur early in childhood development in comorbidity with ADHD [17,18]. It is important to differentiate the difficulties that may arise regarding academic skills from the presence of a specific disorder (Specific Learning Disorder) or simply general difficulties. Specific Learning Disorder (SLD) is a neurodevelopmental condition with biological origins that mainly impacts academic skills, such as writing, reading, and mathematical abilities. It typically emerges during the school years, even when individuals possess adequate intellectual capabilities [19,20]. Several studies focusing on school-aged children with both ADHD and SLD have revealed neuropsychological challenges across various domains, particularly executive functions [21,22]. Specifically, children with ADHD demonstrate significant difficulties in response inhibition and working memory [23,24,25,26,27] while those with SLD exhibit deficits in central executive functioning [28,29], especially concerning working memory [30]. In recent years, the role of inhibition and shifting has garnered increased attention [31]. Nevertheless, only a limited number of studies have thoroughly examined the distinct neuropsychological profiles of children with ADHD and SLD. For instance, some authors [32] demonstrated that individuals with reading difficulties displayed more pronounced impairments in rapid automated naming, phonemic awareness, working memory, and verbal reasoning when compared to those with ADHD. Another study [33] identified specific impairments in impulse control and inhibition among children with ADHD while those with SLD exhibited deficits in phonological level, verbal short-term memory span, and verbal IQ. Studies focusing on graphomotor functioning in children with ADHD have consistently reported that their handwriting tends to be illegible, error-prone, and less organized compared to children without ADHD. These handwriting difficulties, in turn, have a negative impact on their academic performance [34,35]. Researchers have employed digitizing technology, such as computer tablets, along with kinematic analysis of handwriting movements. The kinematic analysis involves examining various variables, such as time, acceleration, speed, and their derivatives, to provide objective and quantifiable markers for investigating handwriting in individuals with ADHD [36]. This form of analysis is particularly relevant in graphomotor research concerning individuals with ADHD, as their poor handwriting is not solely attributed to visual–perceptual or linguistic challenges. Instead, it is believed to be linked to fundamental aspects, such as the regulation of force, speed, and size of movements; motor control; and timing of movements [37,38]. By using digitizing technology and kinematic analysis, researchers can gain deeper insights into the underlying motor control issues contributing to the handwriting difficulties experienced by children with ADHD. Aligned with this interpretation, certain researchers propose that individuals, both children and adults, with ADHD might encounter delays in procedural learning and the automatization of skills, such as handwriting and reading. These skills require sustained attentional control and dedicated practice over an extended period [39]. There is evidence to support this perspective. In a particular study, researchers utilized a serial motor sequence learning task to explore implicit learning in children with ADHD [40]. The results indicated that children with ADHD exhibited a distinct rate of learning compared to the control group while performing motor sequences. Specifically, they demonstrated reduced priming effects, and these differences could not be attributed to poor perceptual-motor abilities. To address the symptoms and impairments experienced by elementary school students with ADHD, three distinct intervention types have been implemented: behavioral strategies, self-regulatory strategies, and academic strategies [41]. Behavioral interventions aim to replace socially undesirable behaviors (e.g., shouting) with socially appropriate behaviors (e.g., working quietly). From a behavioral perspective, each behavior must be understood within the context of its antecedents and consequences [42]. Self-regulation interventions aim to enhance student’s ability to exercise self-control in environments where they experience functional impairments. These interventions typically involve teaching students to identify and record a target response, potentially reinforcing improved performance or accurate recording [43]. In contrast to behavioral and self-regulatory interventions that target ADHD symptoms, academic intervention strategies focus on addressing the functional impairments associated with symptoms. Specifically, these interventions aim to enhance the development of literacy and numeracy skills and improve academic task performance [41]. More precisely, an innovative pedagogical measure that could be beneficial in this context is Universal Design for Learning (UDL). UDL is an instructional design framework developed by the Centre for Applied Special Technology (CAST) that aims to create an educational environment that is accessible and beneficial to every student, regardless of their individual learning needs or disabilities. By incorporating UDL principles, educators can tailor their teaching methods and materials to accommodate a wide range of learners, promoting equal access to knowledge and fostering student engagement and success [44]. UDL is an educational approach that focuses on adapting and flexibly accommodating education and communication to cater to a wide range of cognitive profiles, cultural backgrounds, and sensory functioning. It is a theoretical framework that strives to design curricula that meet the needs of students from the outset of their learning journey and make the education system more inclusive and individualized [45]. Universal Design for Learning (UDL) originated from the concept of Universal Design in architecture (UD), which focuses on designing buildings and infrastructure to ensure access for all individuals, including those with disabilities [46]. In the context of education, UDL represents an approach that aims to enhance access, participation, and progress for all students, including those with disabilities, within the general curriculum [47]. The UDL framework is grounded in contemporary neuroscience research and current learning theories, utilizing digital technology to create a flexible education environment that removes barriers for learners [45]. Neuroscience reveals that each human mind is unique, with distinct differences in anatomy, chemistry, and physiology. Learning occurs through intricate communication within and between three primary networks in the brain: affective networks related to motivation and engagement, recognition networks contributing to information perception and knowledge transformation, and strategic networks involved in planning and organizing procedures and skills [48]. This aligns with educational theorists such as Vygotsky and Bloom, who identified three domains of learning: emotional, cognitive, and skills. The UDL framework provides a customizable structure to design educational programs that cater to the diverse needs of individual learners [49]. It is guided by three core principles: engagement, representation, and action and expression. The engagement principle focuses on strategies to increase student motivation by providing meaningful choices, promoting self-regulation, and offering diverse forms of feedback, including self-assessment and mastery-oriented feedback. Representation involves presenting information in various ways, making it accessible to all students through multiple media, languages, and pedagogical approaches that prioritize checking understanding and message clarity. The action and expression principle emphasizes alternative methods for students to acquire and practice knowledge and skills, utilizing assistive technologies, varied control schemes, and flexible teaching approaches to accommodate individual student needs, such as adjusting text speed or implementing alternative strategies for task completion. Globally, UNICEF [50] has identified the need for better training of educators to include students with disabilities in classrooms, highlighting the necessity of training educators in effective frameworks and practices proven to be effective in teaching students with disabilities in inclusive settings, including UDL. Therefore, it is crucial to translate and disseminate UDL principles on a global scale to support inclusive practices. Already, countries such as Canada, Australia, South Korea, and Portugal have embraced the UDL principles and guidelines, demonstrating its growing acceptance and implementation worldwide. The CAST Institute has translated these principles and guidelines into various languages, including Arabic, Chinese, and others, making them accessible on their website. Given the heterogeneity of studies and the multimodal nature of interventions that can be conducted for individuals with ADHD, exploring the potential effects of an educational model based on general principles that can be adapted to the skills and characteristics of each student is of great interest. Can an innovative educational approach such as Universal Design for Learning be more efficient than traditional didactics concerning the automation of reading, writing, and calculation skills in children with ADHD? Ultimately, this study aims to compare two teaching strategies (traditional inclusive educational strategies and Universal Design for Learning) among a sample of 3rd-grade elementary school children with ADHD and examine the effects of these approaches on their academic and learning abilities.

## 2. Materials and Methods

### 2.1. Participants

In this study, we examined a group of 60 individuals diagnosed with Attention Deficit–Hyperactivity Disorder (ADHD) starting from the third year of primary school. These subjects were divided into two groups, each consisting of 30 individuals. All participants were selected from five primary schools in Caserta, Italy, and came from similar socio-cultural backgrounds in terms of their parents. The family and environmental context did not influence educational attainment in either group. The inclusion criteria for the study were as follows: (a) belonging to the same class level (third grade), (b) having a diagnosis of ADHD without any additional comorbidities as determined by the Kiddie Schedule for Affective Disorders and Schizophrenia (K-SADS) [51], (c) possessing an IQ ranging from 95 to 105 as assessed by the Wechsler Intelligence Scale for Children (WISC-IV) [52,53], and (d) belonging to a medium–high socio-cultural class, as evaluated by the SES scale [54]. After confirming the inclusion criteria for the sample, the subjects were randomly assigned to two experimental groups, each consisting of 30 individuals. Both groups shared the same inclusion criteria and did not differ significantly in terms of socio-cultural factors. The first experimental group (GR2) comprised 30 subjects with an average age of 8.32 (SD 0.43) and an average SES index of 7.10 (SD 0.62), including 14 males and 16 females. The control group (GR1) consisted of 30 subjects with an average age of 8.29 (SD 0.39) and an average SES index of 7.21 (SD 0.68), including 17 males and 13 females. More detailed information can be found in Table 1. To assess academic skills, we employed the following tests: Test MT, Battery for the Evaluation of Writing and Spelling Skills (BVSCO-2), and Test AC-MT 6-11. These tests were administered twice: the first assessment (T0) took place four months after the start of the school year, and the second assessment (T1) was conducted at the end of the school year. Additionally, we used the Wechsler Intelligence Scale for Children (WISC-IV) to assess the participants’ IQ levels [52]. Teachers also completed the Child Behavior Checklist to evaluate the emotional–behavioral profile of the participants.

After the initial assessment, both groups received two different types of interventions, lasting for five months. The data were collected and analyzed at the FINDS Neuropsychiatry Outpatient Clinic by licensed psychologists in collaboration with the Rome University of International Studies (UNINT).

As shown in Table 1, the two groups are comparable in terms of gender, age, socioeconomic background, absence of psychopathologies, and IQ.

### 2.2. Instruments

The protocol used consists of the following tests:

*SES*: a self-administered questionnaire was utilized to gather information regarding the educational level and profession of the parents. This questionnaire also indicated the individual or family’s position within the social system [54].

*WISC-IV*: The participants’ IQ levels were assessed using the Wechsler Intelligence Scales [52,53]. These scales are comprehensive intelligence assessment tools that provide a synthesized measure of intellectual ability. They generate various indices to assess different aspects of cognitive functioning:*Global IQ Index:* This index represents an overall measure of general intellectual ability, taking into account various cognitive skills and abilities;*Verbal Comprehension Index (ICV):* This index assesses verbal reasoning ability, including comprehension and verbal expression based on previously acquired information;*Visuo-Perceptive Reasoning Index (IRP):* This index evaluates perceptive reasoning abilities, such as the ability to analyze visual information and solve visual puzzles;*Index of Working Memory (IML):* This index measures the capacity to retain and utilize information within short periods. It involves tasks that require holding and manipulating information in memory;*Processing Speed Index (IVE):* This index assesses the speed at which individuals can process information efficiently, particularly in timed tasks.

These indices provide a comprehensive and detailed profile of the participants’ cognitive abilities, allowing for a better understanding of their intellectual strengths and weaknesses. The Wechsler Intelligence Scales are widely used in clinical and educational settings to assess cognitive functioning and help tailor interventions and support based on individual needs.

*CBCL*: to evaluate the emotional-behavioral profile, the Child Behavior Checklist was administered, and completed by the teachers. This checklist assesses the profile of children and adolescents, specifically those aged between 1 ½–5 and 6–18, based on the diagnostic criteria outlined in the DSM-IV [55].

*Test MT-3 Clinic*: to measure reading and comprehension skills from primary school to the second year of secondary school [56], the Test MT-3 Clinic was administered.

*BVSCO-2*: to evaluate all aspects involved in the learning path of writing: graphism, spelling competence and the production of written text [57], the BVSCO-2 was administered.

*AC-MT 6-11*: for the assessment of numerical and computational skills including tests of calculation and writing and recovery of arithmetic facts [58], the AC-MT 6-11 was administered.

### 2.3. Procedures

After a period of 4 months from the commencement of the school year, we conducted assessments of the student’s academic skills and emotional profiles using the MT-3 Clinic, BVSCO-2, AC-MT 6-11 tests, and the CBCL. Following this initial assessment, interventions were provided to the two groups as follows: the control group (GR1) received a specialized inclusive teaching approach of the traditional type. This approach involved implementing strategies commonly used for managing children with ADHD. These strategies included structured programming of activities in the classroom, combining various behavioral techniques, such as positive reinforcement, extinction, time out, and token economy. Additionally, time and school environment were organized, changes were monitored, and specific objectives were set. The management of classroom spaces and sources of distraction included providing clear and concise instructions. The experimental group (GR2) underwent the Universal Design for Learning, an innovative educational approach that allows for a high level of customization in the teaching proposal. Specifically, it refers to three brain networks that, by integrating with each other, enhance the learning process: the affective network (motivation), the receptive network (retrieval and processing of external information), and the executive network (planning and re-elaboration of outgoing responses). In particular, students in the group managed through UDL were initially assessed through: (a) an analysis of their motivation to study and any affective obstacles in the learning process; (b) an analysis of their learning style and the main channels used for information reception; (c) an analysis of their executive functioning style, including an exploration of inhibitory systems, working memory, and problem-solving abilities. The first two networks were evaluated through structured and targeted interviews while the third network was assessed through the administration of the Raven’s Matrices [58], Corsi Test [59], and Tower of London tasks [60].

Based on the evaluations conducted, personalized educational plans were created, focusing on the three networks, and the same compensatory measures were chosen according to the assessment results. The study objectives remained the same as those of the class; the only interventions made were based on the evaluation of the affective network. Finally, study materials were selected in line with the analysis of the receptive network.

These interventions were implemented over a period of 5 months. After the school year, we conducted a reassessment of the sample’s academic skills and emotional profiles, utilizing the same methods employed in the initial assessment.

## 3. Results

The data analysis was conducted using the SPSS 26.0 statistical software (2019). A significance level of 1% (α < 0.001) was considered significant.

To assess the effectiveness of the intervention, we compared Groups 1 and 2 at two different time points, T0 and T1, to determine if there were improvements within each group over time. Additionally, we compared both groups at T1 to ascertain which educational intervention was more effective. To perform these comparisons, a two-way mixed MANOVA (Multivariate Analysis of Variance) with a 2*2 design was conducted. The factors analyzed were within-group factor, time (T0 and T1), and between-group factor, group (Group 1 and Group 2).

The two independent variables analyzed were time and group while the dependent variables were the MT and AC-MT tests. The results are presented as follows.

For the MT test, the findings are as follows:

The analysis reveals a significant interaction between the three subscales (scale*time*group) [F (1, 58) = 97.907, *p* < 0.001]. This indicates that there are notable interactions among the subscales, time, and type of treatment. Specifically, at T1, both interventions led to significant improvements in the three subscales of the MT tests. However, in Group 2 (G2), there was a more significant improvement in the correctness parameter compared to Group 1 (G1) (Table 2 and Figure 1).

For the AC-MT test, we found:

The analysis reveals a significant interaction between the two subscales (scale*time*group) [F (1, 58) = 63.140, *p* < 0.001]. This indicates that there are notable interactions among the subscales, time, and type of treatment. Specifically, at T1, both interventions led to significant improvements in the two subscales of the AC-MT tests. However, in Group 2 (G2), there was a more significant improvement in the correctness parameter compared to Group 1 (G1) (Table 3 and Figure 2).

Then, we conducted a comparison between Groups 1 and 2 at two different time points, T0 and T1, by individually analyzing the BVSCO and CBCL tests. The aim was to determine which of the two educational interventions was most effective (within-variable time). Additionally, we compared both groups at T1 to assess the differences between them (between-variable group). For this analysis, we performed a two-way mixed-design univariate ANOVA with a 2*2 design. The factors analyzed were the within-group factor, time (T0 and T1), and between-group factor, group (Group 1 and Group 2).

For the BVSCO test, we found:

The analysis reveals a significant interaction between time and group (time*group) [F (1, 58) = 53.578, *p* < 0.001]. This indicates that there are notable interactions between the two variables, time, and the type of intervention. Specifically, both treatments resulted in an improvement in the BVSCO scale. However, this improvement is more significant in Group 2 compared to Group 1 (Table 4 and Figure 3).

For the CBCL test, we found:

The analysis reveals a significant interaction between time and group (time*group) [F (1, 58) = 178.112, *p* < 0.001]. This indicates that there are notable interactions between the two variables, time, and the type of intervention. Specifically, both treatments resulted in a reduction of externalizing symptomatology in the CBCL scale. However, this reduction is more significant in Group 2 compared to Group 1 (Table 5 and Figure 4).

## 4. Discussion

The significance of a multimodal approach in the treatment of ADHD cannot be overstated. While psychostimulant medications are often the first-line treatment for ADHD and can lead to substantial symptom reduction, it is essential to acknowledge that medication alone is not a comprehensive intervention [61,62,63]. Some parents may have reservations about medicating their children due to concerns about stigma, side effects, or potential long-term effects [64]. Recent studies have indicated that there are no long-term cognitive benefits of medication, such as improved reaction time or verbal working memory, after six years when participants are unmedicated during testing [65]. Therefore, it is widely recognized that a multimodal approach is necessary for effective ADHD treatment. Psychiatric guidelines recommend the incorporation of social skill training, behavioral interventions, and educational approaches alongside medication [66,67,68]. By integrating various strategies, treatment becomes multi-contextual, enhancing benefits and improving the prognosis for individuals with ADHD. The implementation of Universal Design for Learning (UDL) in children with ADHD holds significant potential for addressing the unique challenges they face in the learning environment. UDL is an inclusive educational framework that aims to cater to the diverse needs of all learners, including those with disabilities or learning differences. Children with ADHD often struggle with sustaining attention, managing impulsivity, and organizing tasks, which can impact their academic performance and overall learning experience. UDL’s core principles of providing multiple means of representation, engagement, and expression can be particularly beneficial for these students. In terms of representation, UDL encourages educators to present information in diverse ways, utilizing various media, visual aids, and technology to accommodate different learning styles. For children with ADHD, this means accessing information through visual cues, videos, or interactive materials, which can enhance their comprehension and engagement with the content. The principle of engagement emphasizes providing meaningful choices and promoting self-regulation to increase student motivation. For children with ADHD, this could involve offering them options in how they demonstrate their understanding of a topic or allowing them to choose topics of personal interest for projects. In terms of expression, UDL advocates for flexible methods for students to acquire and demonstrate knowledge and skills. For children with ADHD, this means allowing them to use alternative means of expression, such as oral presentations, graphic organizers, or digital tools, which can better align with their strengths and abilities.

By adopting UDL principles, educators can create a more inclusive learning environment that addresses the specific needs of students with ADHD, promoting their academic success and overall well-being. A multimodal approach that incorporates UDL alongside other interventions can optimize the outcomes for children with ADHD and help them reach their full potential. In our study, we focused on the educational intervention and investigated the impact of Universal Design for Learning as an individualized approach that can address the specific needs of children with ADHD in comparison to traditional educational strategies commonly employed in schools.

We found that both interventions led to improvements in test performance, indicating that interventions were necessary to enhance reading, writing, and arithmetic skills. Furthermore, the group that received an educational intervention based on Universal Design for Learning demonstrated a more significant improvement in these areas. Additionally, we propose that the set of techniques implemented by teachers in the classroom helped children to read, write, and perform math tasks correctly and more fluently. Regarding engagement, UDL promotes creating a positive and inclusive learning environment where students with ADHD feel motivated and supported. Offering choices and opportunities for self-regulation can empower these learners to actively participate in the learning process and foster a sense of autonomy.

We attribute our results to the improved self-regulation stimulated by the individualized approach as supported by previous studies [69]. Over the past decade, numerous studies have provided evidence for the hypothesis that children with ADHD exhibit a generalized deficit in self-regulation, impacting abilities such as information processing, response inhibition, arousal, alertness, planning, executive functioning, metacognition, and self-monitoring [70,71,72,73,74]. As the results suggest, teachers’ knowledge about ADHD and its management techniques is crucial, and a deeper understanding of the cognitive and affective functioning of students may enhance improvements in academic foundational skills within the educational system. Furthermore, it is important to highlight the positive effects reported by teachers. Following the implementation of the intervention, teachers observed improvements in primary symptoms and behavioral difficulties typically associated with ADHD, such as externalizing symptoms. They identified a reduction in hyperactive/impulsive behaviors and a significant increase in self-control among the experimental group. Similarly, in the group that received the traditional intervention, typical problems associated with the disorder were also reduced according to the teachers’ observations. Finally, the collaboration between teachers, parents, and specialists is essential in implementing UDL effectively for children with ADHD. By working together, they can identify the most appropriate accommodations and support strategies, ensuring that the learning environment is conducive to their success.

## 5. Conclusions

In conclusion, this study aimed to compare two commonly used educational interventions for managing ADHD in school settings. The intervention employing traditional personalized teaching strategies demonstrated a positive effect on children’s basic learning skills, underscoring the importance of early intervention. However, Universal Design for Learning proved to be more effective in fostering these abilities. This highlights the value of personalized teaching interventions that consider individual differences beyond mere symptomatology.

### Limitations

Finally, we acknowledge the limitation of our study and emphasize the importance of conducting follow-up assessments to verify the stability of the observed improvements over time. While UDL offers a flexible and inclusive approach to teaching and learning, it does present certain limitations and challenges that warrant consideration:*Complex Implementation:* Fully integrating UDL principles into curriculum design and instruction requires significant changes in educational practice. This may necessitate time, resources, and adequate training for educators;*Access to Technological Resources:* UDL encourages the use of digital technologies to provide multiple modes of content presentation. However, not all schools or districts have the resources to offer a wide range of technologies or reliable internet connections;*Measurement and Assessment:* Assessing student performance within the UDL model can be more complex compared to traditional assessment methods. Measuring student progress through flexible and personalized assessments can pose challenges;*Lack of Experimental Evidence:* Despite UDL’s strong theoretical foundations and research basis, there may still be a lack of long-term experimental evidence to demonstrate its effectiveness in every educational context;*Financial Resources:* Full implementation of UDL may require significant financial investments for acquiring resources, technologies, and educator training.

Despite these limitations, UDL remains a valuable approach to creating an inclusive and flexible learning environment that can benefit many students with diverse needs. However, it is important to continue scientific research using the UDL to provide additional data and develop a more solid scientific foundation for this educational methodology, especially considering the scarcity of studies.

## Figures and Tables

**Figure 1 children-10-01350-f001:**
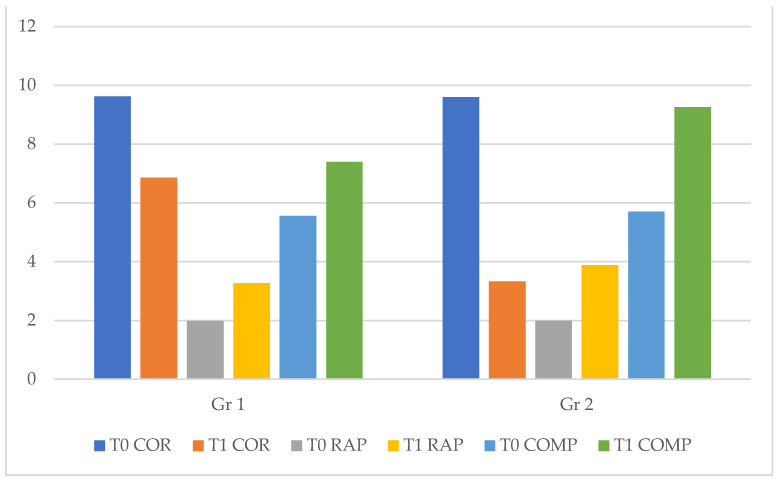
Interaction between scale*time*group in the MT tests.

**Figure 2 children-10-01350-f002:**
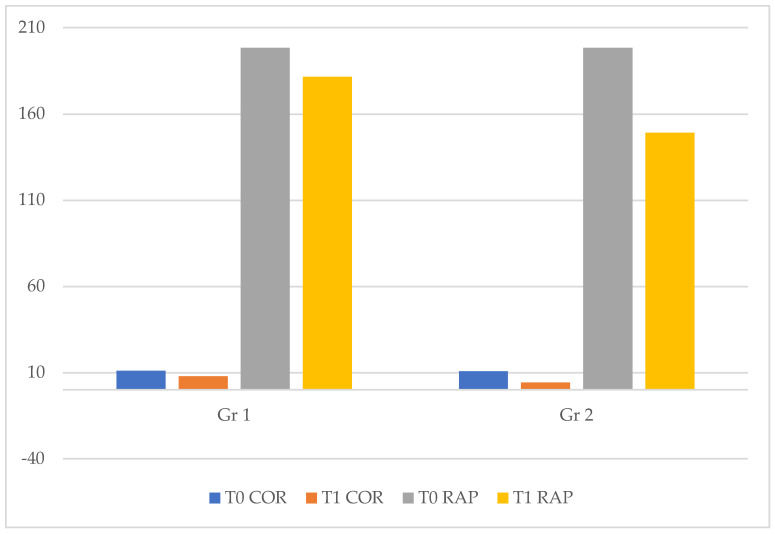
Interaction between scale*time*group in the AC-MT tests.

**Figure 3 children-10-01350-f003:**
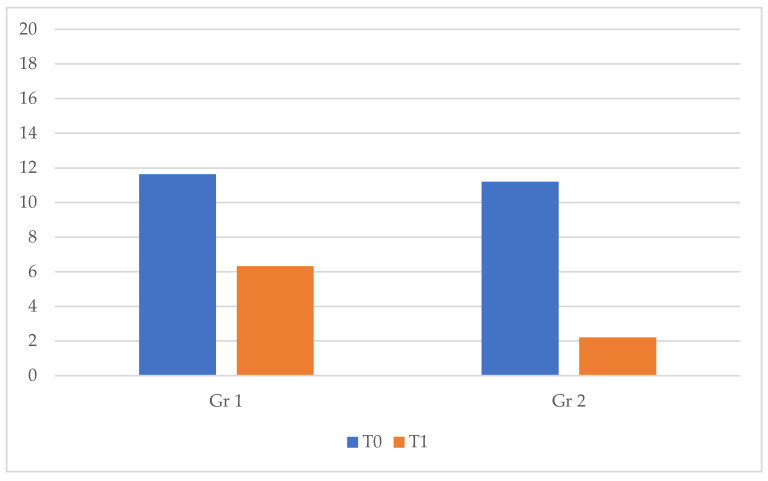
Interaction between time*group in the BVSCO scale.

**Figure 4 children-10-01350-f004:**
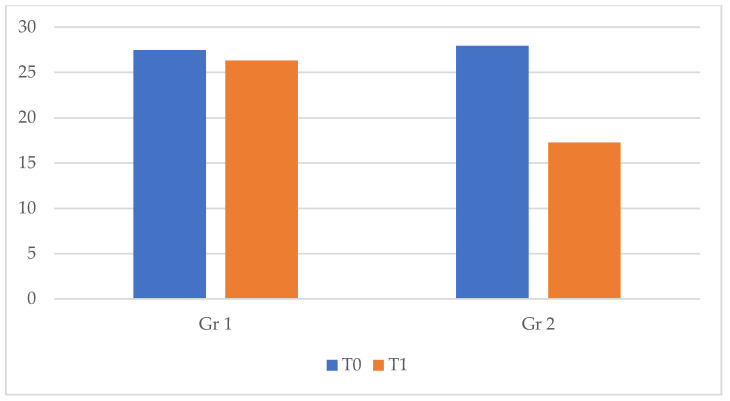
Interaction between time*group in the CBCL scale.

**Table 1 children-10-01350-t001:** Description of the sample.

	All	Gr1	Gr2
N	60	30	30
Male (number)	31	17	14
Age (years)	8.30 (0.39)	8.29 (0.39)	8.32 (0.43)
Sociocultural background (SES)	7.15 (0.60)	7.21 (0.68)	7.10 (0.62)
Comorbidities (K-SADS)	Absence	Absence	Absence
Intelligence (WISC-IV)	97.46 (1.08)	96.48 (1.09)	98.44 (1.12)

**Table 2 children-10-01350-t002:** Interaction between scale*time*group in the MT tests.

Group	MT	Time	Means	SD	F	*p*
1	Accuracy	T0	9.63	1.15		
		T1	6.86	1.33		
	Fluency	T0	1.98	0.11		
		T1	3.27	0.36		
	Comprehension	T0	5.56	0.67		
		T1	7.40	0.67		
2	Accuracy	T0	9.60	1.32		
		T1	3.33	0.95	97.907	<0.001 *
	Fluency	T0	1.99	0.10		
		T1	3.89	0.21		
	Comprehension	T0	5.70	0.74		
		T1	9.26	0.90		

* Statistical significance.

**Table 3 children-10-01350-t003:** Interaction between scale*time*group in the AC-MT tests.

Group	AC-MT	Time	Means	SD	F	*p*
1	Accuracy	T0	11.00	1.20		
		T1	7.96	0.76		
	Fluency	T0	198.40	14.33		
		T1	181.63	7.59		
2	Accuracy	T0	10.76	1.33		
		T1	4.26	0.69	63.140	<0.001 *
	Fluency	T0	198.20	12.57		
		T1	149.13	4.64		

* Statistical significance.

**Table 4 children-10-01350-t004:** Interaction between time*group in the BVSCO scale.

Group	Time	Means	SD	F	*p*
1	T0	11.63	1.42		
	T1	6.33	0.84		
2	T0	11.20	1.18		
	T1	2.20	1.42	53.578	<0.001 *

* Statistical significance.

**Table 5 children-10-01350-t005:** Interaction between time*group in the CBCL scale.

Group	Time	Means	SD	F	*p*
1	T0	27.46	2.54		
	T1	26.30	3.32		
2	T0	27.93	2.09		
	T1	17.26	2.54	178.112	<0.001 *

* Statistical significance.

## Data Availability

The data presented in this study are available on request from the corresponding author.

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
