# Peer review of "Universal Design for Learning for Children with ADHD"

_children, 2023, doi:10.3390/children10081350_

Round 1

Reviewer 1 Report

The paper reports research on children with ADHD, which is related to symptoms such as learning difficulty, motor impairment, anxiety or depressive disorders. It focused on a specific, innovative educational approach that could improve basics learning skills to manage the issues that could occur with ADHD. The program followed, namely the Universal Design for Learning, is an individualized approach that combines current neuroscientific knowledge creating a personalized teaching on the strengths and weaknesses of the student. The application of this program was accompanied by an assessment procedure and the results are presented as research findings which suggests that the interventions led to improvements in test performance. On these results, the authors propose that the techniques implemented by teachers in the classroom helped children improve reading, writing and math skills.

 The paper reports ah interesting intervention that included measurement of skill via valid instrumentation, while the empirical part is correctly performed and article is overall well written.

Some minor revision is needed, and additional issues should be addressed in the methodological section, which will improve the paper.

 - Please provide reliability measures for instruments used.

 -Make clear how the selection of the two groups is made, and account for their equivalence.

 -Graph presenting the relations of time*group, it is suggested to use, so the changes could be visualized and easily understood.

- Present a separate section discussing the limitations of the method.

Minor editing of English language required

Author Response

Dear reviewer,

My colleagues and I deeply appreciate the corrections and the time you have dedicated to reading and revising the manuscript. We are delighted that you found it interesting and well-done. Despite this, we have added sections that address your comments, and in the attached file, we have attempted to provide responses to the issues you raised. We hope that the revised manuscript will be satisfactory. Thank you for the support and assistance you have provided.

Kind Regards

Reviewer 2 Report

The study presented addresses a problem of interest and is relevant to the profile of the journal.

The writing has a good structure and its writing is clear and precise.

A good statement of the problem is made, but it is suggested to highlight the research questions in the introduction section, before presenting the objectives of the study.

Regarding the procedures, it is suggested to provide more background on both interventions and the fundamentals of their design.

It is important to mention the ethical protocols to ensure the participation of the subjects.

It is advisable to develop a broader discussion of the results obtained, deepen the limitations of the study and declare the contributions of the results for decision-making by educational centers and educational professionals who work with students with this type of diagnosis.

Finally, it is suggested to develop some ideas of projections of the study or new research questions that remain open from the study carried out.

Author Response

(The authors gave the same response as above.)
